# Aiding Cancer’s “Sweet Tooth”: Role of Hexokinases in Metabolic Reprogramming

**DOI:** 10.3390/life13040946

**Published:** 2023-04-04

**Authors:** Zeenat Farooq, Hagar Ismail, Sheraz Ahmad Bhat, Brian T. Layden, Md. Wasim Khan

**Affiliations:** 1Division of Endocrinology, Diabetes, and Metabolism, Department of Medicine, The University of Illinois at Chicago, Chicago, IL 60612, USA; 2Jesse Brown Veterans Affairs Medical Center, Chicago, IL 60612, USA

**Keywords:** cancer metabolism, HKDC1, hexokinases, glucose metabolism, metabolic reprogramming

## Abstract

Hexokinases (HKs) convert hexose sugars to hexose-6-phosphate, thus trapping them inside cells to meet the synthetic and energetic demands. HKs participate in various standard and altered physiological processes, including cancer, primarily through the reprogramming of cellular metabolism. Four canonical HKs have been identified with different expression patterns across tissues. HKs 1–3 play a role in glucose utilization, whereas HK 4 (glucokinase, GCK) also acts as a glucose sensor. Recently, a novel fifth HK, hexokinase domain containing 1 (HKDC1), has been identified, which plays a role in whole-body glucose utilization and insulin sensitivity. Beyond the metabolic functions, HKDC1 is differentially expressed in many forms of human cancer. This review focuses on the role of HKs, particularly HKDC1, in metabolic reprogramming and cancer progression.

## 1. Introduction

First observed by Otto Warburg in 1924, one of the hallmarks of cancer cells is reprogrammed glucose metabolism, where glucose uptake and lactate production are enhanced regardless of oxygen concentrations, popularly known as the “Warburg effect.” Initially, this phenomenon was thought to be due to mitochondrial dysfunction in cancer cells [1]. However, research has now established that enhanced glucose metabolism coupled with altered mitochondrial metabolism supplies increased energy needs and provides metabolites for biosynthetic pathways, such as nucleotides, fatty acids, and amino acids needed by proliferating cancer cells [2]. The first step of glucose metabolism is catalyzed by hexokinases (HKs). HKs are a family of phosphotransferase enzymes with different kinetic properties, expression profiles, and subcellular localization that initiate glucose metabolism [3,4,5]. Four canonical isoforms of the HK family have been well characterized: HKs 1–3 have a broad range of expression, and the fourth isoform, more commonly known as glucokinase (GCK), is expressed mainly in the liver and pancreas [3,4,5,6]. Although specific roles have been described for each HK, the existence of multiple isozymes catalyzing the same reaction within the same cell or tissue is a pressing question. Glucose-6-phosphate (G6P) is the first stable intracellular intermediate of glucose metabolism; therefore, its generation is tightly regulated by the selective expression of different HK isoforms in normal and pathophysiological scenarios [7]. For the same reason, HKs vary in cellular distribution, expression patterns, and substrate affinity levels depending on the cell’s physiological state. This review describes the role, distribution, and regulation of different isoforms and their metabolic functions in cancer.

### 1.1. General Characteristics and Distribution

Genes that code for HK protein isoforms are conserved from bacteria to humans [3,5]. However, bacterial and lower vertebrate genes code for smaller proteins (about 50 kDa), whereas mammalian HKs 1-3 and HKDC1 are ≅100 kDa in size. HK isoforms possess high sequence similarities (Appendix A) at the ‘N’ and ‘C’ terminal domains referred to as “Hemi domains” (Figure 1) [3,4,5,8,9,10,11,12,13,14,15,16,17,18,19,20,21]. These hemi-domains are thought to have evolved because of a gene duplication event from the bacterial HK enzyme [22,23]. Upon subsequent evolutionary divergence, the N-terminal hemi-domain acquired different properties in each isoform (Figure 1) [8,9,10,11,12,13,14,15,16,17,18,19,20,21,24,25]. One characteristic feature of mammalian HKs is allosteric inhibition by G6P, which supports the “gene duplication event theory,” suggesting that the duplication event led to the formation of a hemi-domain that evolved into a regulatory binding site [26]. Amino acid sequence comparisons of hexokinases from lower to higher vertebrates are available to support the gene duplication hypothesis (Appendix A). Various comparison analysis studies suggest that high similarities in sequence exist not only between the hemi-domains but HKs from lower to higher vertebrates, indicating a slow rate of amino acid substitution (rate of mutation through evolution) at homologous HK genes across species [26,27,28]. Some of the common characteristics of each isozyme are described and listed in Table 1. The HK1 gene encodes a protein of 100 kDa, and only the C-terminal domain is catalytically active [3,5]. It is ubiquitously expressed inside all cells with a granular cytoplasmic expression pattern. The enzyme is also localized to the mitochondrial outer membrane [3,4,5]. HK1 has the highest level of tissue expression in the brain, followed by the urinary bladder, thyroid gland, colon, and bone marrow [3,5]. HK2 is the most well-characterized isoform of the HK family, primarily expressed in insulin-sensitive tissues such as adipose and skeletal muscle [29]. It undergoes significant changes in expression in different cancers and is the most well-studied HK in cancer biology [30,31,32,33,34,35,36,37,38,39,40]. It is the only identified HK with both N and C terminal domains catalytically active and is the most highly regulated isoform [3,4,5,9,10,11]. Like HK1, HK2 has been shown to localize to mitochondria [5]. HK3 is a less well-characterized 100 kDa isoform of the hexokinase family, which lacks the N-terminal mitochondrial binding domain of HK1 and 2 (Figure 1). HK3 is expressed in lung, kidney, and liver tissues at lower levels than HK1 and 2. It is also the predominant isozyme in granulocytes [10,11,41] (Table 1). HK4, or glucokinase (GCK), is a unique 50 kDa enzyme mainly expressed in the liver and pancreas and closely resembling the ancestral bacterial enzyme [42,43]. The enzyme is also expressed in enteroendocrine cells and the brain [44,45]. The distinguishing feature of GCK in metabolic regulation is its role as the body’s primary glucose sensor. Small fluctuations in GCK activity alter the threshold for glucose-stimulated insulin secretion (GSIS) from pancreatic β-cells, which is not observed with other hexokinases [11,28,46]. Mutations in the GCK gene lead to two different diseases of blood glucose regulation: maturity-onset diabetes of the young type 2 (MODY-2), and persistent hyperinsulinemic hypoglycemia of infancy (PHHI) [47,48,49]. GCK is localized in the cytoplasm, but reports have also suggested that GCK forms a heteropentameric complex at the mitochondria with BCL 2-associated death promoter (BAD), protein kinase A (PKA, cAMP-dependent protein kinase), protein phosphatase 1 (PP1, dual-specificity serine/threonine phosphatase), and Wiskott–Aldrich family member (WAVE1) under certain conditions to integrate glycolysis and apoptosis [50,51].

Phylogenetic analyses carried out in the middle of the 2000s to comprehend the diversification of the HKs and the evolution of GCK [19,20] led to the discovery of a novel HK gene known as the hexokinase domain containing-1 (HKDC1). The gene that codes for HKDC1 lies on chromosome 10 in humans near the HK1 gene, and the two share more than 70% sequence similarity [20]. The enzyme has been shown to play a role in modulating glucose tolerance during pregnancy by identifying its genetic variants in a genome-wide association study (GWAS) [13,18]. Like HK1-2, it also contains an ‘N’ and a ‘C’ terminal domain, which are the regions predicted to bind glucose and ATP, respectively. It includes amino acid residues, which remain conserved with those of the other HKs (Figure 1). [19,20]. HKDC1 is broadly expressed in the retina of the eyes, kidneys, brain, small intestine, duodenum, pharynx, esophagus, and thyroid gland [20]. We [14,16,52] and others [53] have shown that HKDC1, like HK1-2, associates with the outer mitochondria membrane via its interaction with the voltage-dependent anion channel (VDAC). 

### 1.2. Regulation of Hexokinase Expression

The primary metabolic role of HKs is the phosphorylation of glucose, thus trapping it inside cells and initiating glucose metabolism [3,4,5,54]. HKs, therefore, dictate the direction of glucose flux within the cells. However, if the phosphorylation of glucose were the only role of HKs, the presence of a single HK would seem reasonable. The existence of different isoforms raises a question about the non-redundant role played by each enzyme. It also suggests that different rates of glucose phosphorylation are needed depending on the cell/tissue requirements. This allows “metabolic plasticity” to the cells to allow better regulation and channeling of glucose metabolism, and the role of GCK in this context has been defined most elaborately [6]. Therefore, the expression of HKs is profoundly altered in cancer cells, and it varies widely among different cancer types (Appendix A).

Transcriptional Control: This can be illustrated by highlighting differences in transcriptional regulation between different isoforms. Promoter regions of different hexokinases have been analyzed to contain several regulatory elements governing different transcription profiles under varied conditions [55]. The isoforms have also been observed to bind multiple transcriptional factors [56,57,58,59,60,61,62,63]. The promoter of HK1 in rats has been shown to contain transcriptional start site elements, lack a TATA sequence, and lie within a CpG island that extends into the translational start site [64]. These features are similar to promoter element features of housekeeping genes, tailored for their ubiquitous expression [64]. HK1 promoter also contains regulatory sites known as sp sites within the P2 BOX, which are essential for promoter activity and the binding of protein factors in lower vertebrates to humans [57,65,66]. The role of non-coding elements in regulating HK1 and their association with congenital hyperinsulinism has also been reported [65]. Alternative splicing generates multiple HK1 isoforms, i.e., HK1 (ubiquitous), HKR (erythrocytes), HK-TA/TB (testis-A/B), HK-TB (testis-B), and HK-TD (testis-D). HK1 and HKR isoforms differ only in exon 1 and share the remaining 17 exons, whereas HK-TA, HK-TB, HK-TC, and HK-TD have different 5′ UTR exons share the 17 exons with all other isoforms of HK1 [64]. On the other hand, the promoter region of HK2 contains a single transcriptional start insulin-binding element, leading to the transcriptional upregulation of HK2 by insulin [56]. However, the promoter of HK2 contains a binding motif for hypoxia-inducible factor 1α (HIF1α). HIF1α is upregulated due to hypoxia brought about by the tumor microenvironment, which results in the upregulation of HK2, making HK2 the most highly expressed HK in multiple tumors. Recently, it has been reported that HIF1α is negatively regulated by the long non-coding RNA LINC00365 in breast cancer cell lines, resulting in a decline in HK2 levels and cellular proliferation [67]. The enhanced expression of HK2 in hepatocellular carcinoma in rat models has also been reported to be induced by a loss of DNA methylation on the CpG island on the HK2 promoter [68]. The post-transcriptional regulation of HK2 activity by MicroRNAs has also been documented. Anti-tumorigenic microRNA miR 143 acts on HK2 mRNA, leading to its degradation and decreased stability. The upregulation of oncogenic microRNA miR155 occurs in multiple tumors, repressing miR 143, thereby stabilizing HK2 mRNA [69,70]. HK3 contains a binding site for the basic leucine zipper transcription factor CCAAT/enhancer binding protein alpha (CEBPA) which leads to its transcriptional upregulation during all-trans retinoic acid (ATRA)-mediated neutrophil differentiation [41]. Interestingly, the HKDC1 promoter contains high levels of epigenetic marks such as H3K4me1 and H3K27ac in multiple human cell lines. The regulatory roles of these marks on their expression in normal and cancer cells need further exploration [21]. 

Most cancers shift their HK expression profiles in favor of HK2, a consequence of metabolic re-programming in cancer. This could be illustrated by the induction of gene expression of HK2 and the silencing of GCK in liver and pancreatic cancers [71,72,73,74,75]. In humans, progression from the normal brain to low-grade gliomas and finally to glioblastoma multiforme (N) occurs with a progressive shift from HK1 to HK2 with a concomitant decrease in prognosis. HK2 expression levels are closely associated with tumor grade and mortality in hepatocellular carcinoma and breast metastasis [30,40]. One of the reasons for this response is the catalytic activity in both domains in HK2, favoring greater utilization of glucose and maintaining a downhill gradient for glucose phosphorylation. Additionally, HK2 expression in cancers is stimulated by the insulin signaling pathway Akt/mTORC1, which is upregulated in tumor cells to regulate glucose metabolism, cellular growth, and survival through the phosphorylation of target molecules. Akt phosphorylates HK2 at Threonine 473, which lies within the Akt consensus-binding motif RARQKT*, stabilizing HK2 protein. This motif is conserved from mice through humans [76]. For the same reason, the expression of HK2 is decreased in type 1 diabetes mellitus (T1DM) due to reduced insulin signaling and is recovered upon insulin treatment in T1DM [56,77,78,79,80]. Although HK1 and HKDC1 share the mitochondrial localization property with HK2, they lack the Akt consensus motif, making them more susceptible to degradation through apoptosis. However, we have previously reported that in mouse models, overexpressing HKDC1 enhances Akt phosphorylation (15,16).

### 1.3. Regulation of Hexokinase Activity

HK isoforms have different catalytic and regulatory properties. HK1 is activated by high inorganic phosphate levels (Pi) and inhibited by the product G6P. Therefore, a cellular milieu with a high ratio of Pi/G6P because of high rates of ATP utilization favors glycolysis through HK1 activity for the generation of ATP [5]. One of the best examples to illustrate this is the reversal of the G6P-induced inhibition of HK1 by inorganic phosphate (Pi), which leads to the evasion of the G6P-induced feedback inhibition of glucose phosphorylation and favors its ubiquitous expression since glycolysis is a primary requirement of all mammalian cells [3,4,5].

On the other hand, HK2 lacks this antagonizing response by Pi, and instead, Pi adds to the inhibition caused by G6P in the case of HK2 [45]. This feature favors HK2 activity in metabolically active tissues such as skeletal muscles to replenish glycogen synthesis following muscle contraction. Existing literature suggests an anabolic role for HK2 [11,45]. Additionally, a wealth of literature agrees with an anabolic role for HK2, funneling G6P to synthesize NADPH for lipid biosynthesis via the pentose phosphate pathway (PPP) in the liver and mammary glands [81,82]. 

HK3 is known to be inhibited by glucose at high concentrations of 1 mmol l^−1^ (substrate inhibition) but is less sensitive to inhibition by G6P. Interestingly, HK3 responds towards G6P and Pi similarly to HK2 (Table 1), which supports an anabolic role for HK3; further research is needed to answer this question [28,45]. It also has the lowest affinity for the second substrate, ATP, among all HKs, but the physiological role of this property remains elusive [45].

GCK has the highest Km (lowest affinity) for glucose among all canonical HKs (HK1-4), allowing the liver and pancreas to serve as a “glucose buffer” and a “glucose sensor,” respectively. It is not inhibited by G6P and has a 50-fold lower affinity for glucose than other isoforms. Within the liver, the low affinity is tailored to ensure the availability of glucose to physiologically sensitive tissues such as the brain under starvation and its utilization only when glucose is abundantly available. Within the pancreas, this feature allows GCK to act as a “*glucose sensor*” to regulate insulin release. Mutations in the glucokinase (GCK) gene lead to maturity-onset diabetes of the young, type 2 (MODY-2), and persistent hyperinsulinemic hypoglycemia of infancy (PHHI) [47,49,83]. MODY-2 is a mild type 2 diabetes resulting from a defect in glucose-induced insulin secretion [47,48,49]. Mutations in the GCK leading to MODY-2 are arguably the most common cause of monogenic diabetes due to these specific mutations. More than 40 mutations have been linked to MODY-2, including frameshifts, nonsense, missense, and splice-site variants [1,2,3,4,5,6,7]. The proposed role of GCK as a “glucose sensor” in pancreatic β-cells [2,11,12] is consistent with the MODY-2 phenotype wherein small reductions in β-cell activity increase the threshold for glucose-induced insulin secretion resulting in the phenotype. However, a report by Postic et al. suggests that hepatic GCK also plays a role in MODY-2. Alterations in GCK activity are also associated with many other diseases that have been reviewed elsewhere in detail [12,42]. Owing to its unique role, GCK regulation is complex, and several regulatory mechanisms have been discovered. Alternative and tissue-specific promoters drive GCK transcription and gene expression to varying degrees [84,85,86,87,88,89]. Several metabolites, including insulin, glucose, and hormones, regulate GCK expression at the transcriptional level [43,90,91,92,93]. The regulation of GCK has been recently reviewed elsewhere in more detail [94].

Not much is known about the kinetic and regulatory properties of HKDC1, and it needs further exploration. However, the genetic locus near HKDC1 is a “hot spot” for various “histone modifications,” and it is believed that HKDC1 is subject to different levels of regulation under different physiological and pathophysiological conditions [18,21]. Although HKDC1 has two kinase domains like HK1, there have been contrasting reports on its catalytic potential. An early study suggests that HKDC1 possesses hexokinase activity, where experiments on INS-1 rat pancreatic cells with HKDC1 overexpression showed changes in HK activity [21]. Interestingly, the hexokinase activity of the other HKs was unaffected by the expression of HKDC1 [21]. Going further, our group has recently shown that the hexokinase activity of HKDC1 is quite low, and the principal function of the protein may be more related to binding to mitochondria and modulating glucose flux [16].

### 1.4. Differences in Subcellular Localization

This feature allows the utilization of different HK isoforms for channeling G6P to pathways dictated by the cell’s metabolic state. Under normal conditions, GCK is primarily cytosolic [95], whereas HK3 is mostly perinuclear in localization [96]. The subcellular localization of HK1 and 2 has important influences on their metabolic, antioxidant, and anti-apoptotic effects. HK1 localizes to the mitochondrial membrane, and HK2 is localized to the outer mitochondrial membrane through a voltage-dependent anion channel (VDAC) [3,5]. However, HK2 binds to mitochondria with less affinity than HK1 and can translocate between cytoplasm and mitochondria depending on glucose and glucose 6-phosphate [7]. Mitochondrial-bound HK1 promotes efficient glucose catabolism by coupling glycolysis with oxidative phosphorylation. This feature makes it the ideal HK isoform for brain cells. On the other hand, HK2 in normal cells is mostly cytosolic and promotes anabolic functions such as glycogen synthesis through PPP, making it ideal for muscle and cardiac cells. Additionally, PPP leads to the generation of reduced glutathione from NADPH which is essential for the antioxidant activity of HK2. Although the mitochondrial binding property of HK2 appears to be in tune with the metabolic demands of cancer cells, allowing them to couple glucose consumption with energy (ATP) production, its role in mediating glucose consumption and anabolic processes under normal conditions remains elusive [3,4,5,30,31,32,33,34,35,36]. Studies, however, show that HK2 dynamically shuttles between the mitochondria and cytoplasm in response to changes in intracellular G6P, pH, and Akt signaling pathways [97].

As a result of low-grade inflammation (aging and diabetes), HK1 has been shown to predominantly localize in the cytoplasm and favor an inflammatory phenotype [98,99]. In a landmark study conducted by De Jesus et al., it was observed that mice lacking the N-terminal mitochondrial binding domain (MBD) on HK1 produced an inflammatory response when challenged with lipopolysaccharide (LPS), increased glucose flux through the PPP but decreased flux below the level of glyceraldehyde phosphate dehydrogenase (GAPDH) brought about by the nitrosylation of GAPDH which leads to reduced GAPDH activity [100]. HK3 has also been shown to be associated with mitochondrial-associated membranes (MAMs) in normal mice brains through unknown mechanisms. This effect is abolished due to chronic stress in mice [101]. It has been reported that hexokinases are differentially translocated within cells depending upon the physiological conditions and the mechanisms through which HKs migrate between cellular compartments; however, they remain unidentified and warrant more investigation in this area [102]. Recently, HKDC1 has also been shown to bind with mitochondria via interaction with VDAC [15,16,17,18]. More research is needed in this area to understand better the significance of the differential localization of hexokinases under different conditions.

### 1.5. Roles of Hexokinases in Cancer-Mediated Metabolic Reprogramming

One of the characteristic features of cancer is unabated cell division. For this reason, neoplastic cells preferentially obtain energy and biomolecules through glycolysis through metabolic reprogramming. Metabolic reprogramming refers to the ability of cancer cells to alter their metabolism to support their enhanced metabolic requirements of high ATP and intermediates for biosynthetic processes. This requirement brings about extensive changes in the expression of different hexokinase enzymes.

*Hexokinase 1:* The expression of HK1 is amplified in some cancers where it is responsible for rewiring the metabolic state towards aerobic glycolysis to supply ATP and macromolecules (Figure 2) [103,104,105]. The observation that most normal cells express HK1 while cancer cells express HK1 and HK2 stimulated interest in reducing HK2 activity in cancers. However, studies have demonstrated that the knockdown of HK2 alone does not inhibit in vivo tumor progression with reduced glucose consumption, suggesting that HK1 compensates for the overall tumorigenic potential. In contrast, the knockdown of HK2 in HK1- HK2+ cancers reduced xenograft tumor progression [106,107,108,109]. These studies suggest a greater involvement of HK1 in tumor progression beyond its currently known role and possibly as a regulatory function in cancer cells. For example, in a study by Daniela et al., HK1 has been shown to be involved in ovarian cancer in a glucose phosphorylation-independent fashion [110]. HK1 also serves as the effector of KRAS4A, an isoform of the most frequently mutated oncogene KRAS, during tumorigenesis [111].

*Hexokinase 2:* HK2 is significantly overexpressed in treatment-resistant primary and metastatic breast cancer [37,38,39,40], bladder cancer [112], cervical squamous cell carcinoma [113], colorectal cancer [114], neuroendocrine tumor [103], ovarian epithelial tumors [104], glioblastoma [55,105], hepatocellular carcinoma [30], laryngeal squamous cell carcinoma [31], lung cancer [32], neuroblastoma [33], pancreatic cancer [34], and prostate cancer. HK2 expression in these cancers inversely correlates to overall patient survival rates [35]. The genetic ablation of HK2 is known to inhibit malignant growth in mouse models [36,106,107,108,109]. A landmark study on an adult tumor model of mice demonstrated the therapeutic effects of systemic deletion of HK2 [36,115,116,117,118]. In addition to its enzymatic activity, the mitochondrial binding ability of HK2 plays a role in inhibiting apoptosis and upregulating synthetic pathways which support tumor growth (Figure 2). The mitochondrial-bound HK2 is therefore elevated in many forms of cancer [37,38,39]. The amplification of HK2 appears to be related to the expression of p53. Recent studies have shown that p53-inducible protein TIGAR (Tp53-induced glycolysis and apoptosis regulator), Akt, and ER stress sensor kinase could regulate mitochondrial HK2 localization [32,117,119,120,121,122,123]. Interestingly, the mitochondrial TIGAR–HK2 complex upregulated HK2 and hypoxia-inducible factor 1 (HIF1) activity, which limits reactive oxygen species (ROS) production and protects against tumor cell death under hypoxic conditions [124,125,126,127,128,129]. It is also observed that the GCK to HK2 switch occurs in hepatocellular carcinoma (HCC), and the expression of HK2 is highest in HCC [127]. Additionally, HK2 is also regulated by epigenetic mediators, including long non-coding RNAs [38,39,106,107], microRNAs [123,124,125,126,127], histone, and DNA methylation [109]. HK2 is localized to the outer mitochondrial membrane through a voltage-dependent anion channel (VDAC) [126] (Figure 2). This association permits direct access to the ATP generated within the mitochondria [124]. This phenomenon is especially significant in malignant cells where rates of aerobic glycolysis go up tremendously to meet the energy demands of the transformed cell (Warburg effect) [105].

*Hexokinase 3:* HK3 is upregulated in several cancers, including acute myeloid leukemia (AML), where it plays the role of an anti-apoptotic protein to promote tumor cell survival alongside HK1 and 2 [130,131]. The previously identified functions of the enzyme include cell survival through the attenuation of apoptosis and the enhancement of mitochondrial biogenesis [2,132,133]. The latest research about the functions of HK3 in normal and cancer cells has uncovered previously unanticipated roles of this protein. A recent study by Seiler et al. has reported that hexokinase 3 enhances myeloid cell survival via non-glycolytic functions [134]. In contrast, another report by Xu et al. showed that HK3 dysfunction promotes tumorigenesis and immune escape by upregulating macrophage infiltration in renal cell carcinoma [135].

*Glucokinase:* Glucose phosphorylation activity for GCK has been observed in several cancer cell lines [136]. GCK is also known to interact with BAD (Bcl-2 agonist of cell death) to integrate glycolysis with apoptosis [50,137,138,139]. To date, 17 activating mutations targeted by multiple activators have been identified in the allosteric activator site of GCK [140,141,142,143]. The activating variations and their targeting by the activators lead to enhanced cellular proliferation, including the proliferation of cancer cell lines such as INS, which indicates a putative pro-oncogenic role for GCK [144,145,146]. Although there is no direct evidence for the role of GCK as a pro-oncogene, recent reports exploring somatic variations of allosterically regulated proteins in cancer genomes suggest that somatic mutations of GCK could play a role in tumorigenesis [147]. Těšínský et al. provide the first direct evidence of the role of GCK in tumorigenesis by demonstrating a change in the kinetic properties of GCK which include an increased affinity for glucose and changes in cooperative binding [148].

*Hexokinase domain containing 1:* Studies performed over the past decade have linked HKDC1 to various functions (Figure 3). Much of the interest in HKDC1’s role in cancer stems from the fact that, like HK1 and 2, it localizes in the mitochondrial outer membrane (MOM) and binds with the voltage-dependent anion channel (VDAC) [14]. We were the first to identify the role of hepatic HKDC1 in glucose metabolism. Using a mouse model of HKDC1, we demonstrated that hepatic HKDC1 modulates glucose metabolism and insulin sensitivity in mice. Although HKDC1 has nominal expression in normal hepatocytes [17], it is significantly upregulated in hepatocellular carcinoma (HCC) cells [149,150], implying that it plays an essential role in HCC. By using HKDC1 knockout models, we have shown that cellular HK activity is not affected by HKDC1 ablation; however, there is a significant increase in glucose uptake, where the bulk of glucose carbons flow through the glycolytic shunt pathways PPP and HBP (Figure 3) [16]. We further show that HKDC1 interacts with the mitochondria, and its loss results in mitochondrial dysfunction [16]. Since cancer cells require ATP to prepare for cell division during the synthetic (S) phase of the cell cycle, a deficiency in ATP may cause cell cycle arrest. Others have shown that HKDC1 is also significantly increased in breast cancer cells, enhancing glucose uptake and mitochondrial membrane potential to encourage cell survival and growth. In agreement with this phenomenon, HKDC1 knockdown increased the production of reactive oxygen species (ROS), the activation of caspase 3, and apoptosis [52]. Li et al. [151] used RNA-seq data from The Cancer Genome Atlas to pinpoint genetically altered genes in a univariate survival analysis of patients with squamous cell lung carcinoma (SQCLC). Seven thousand two hundred twenty-two genetically modified genes were discovered by the analysis of RNA-seq data from 550 SQCLC patients, and HKDC1 was one of 14 feature genes with more than 100 frequencies linked to a worse prognosis [151,152]. HKDC1 mRNA and protein levels were also expressed higher in lung cancer cell lines than in healthy lung epithelial cells.

Additionally, there was a direct correlation between the degree of HKDC1 protein expression and histological differentiation, reduced survival, tumor size, pN (N refers to the number of nearby lymph nodes with cancer) stage, and poor prognosis. In agreement with these results, lung cancer cell lines stably overexpressing HKDC1 demonstrated increased glucose consumption, lactate generation, proliferation, migration, and invasion compared to healthy lung epithelial cells [151,152]. A comparison study on RNA sequencing (RNA-Seq) analysis of colorectal cancer (CRC) and matched standard tissue samples has observed significant splicing variations in nine genes in CRC. Interestingly, the authors discovered alternate regulation of the first exon in HKDC1 using exon sequencing (DEXSeq) to uncover variations in relative exon usage. HKDC1 E1a-E3a was elevated in CRC, suggesting a potential functional impact because of a projected change in the HKDC1 protein sequence [153,154,155]. Another study has reported a 13 h phase change in HKDC1 expression between SW480 cells and their metastatic counterpart SW620 (a core clock gene) that occurs in conjunction with a phase shift in aryl hydrocarbon receptor nuclear translocator-like protein-1 (BMAL1). In SW480 cells, silencing BMAL1 results in an elevation of HKDC1 expression, and this effect was eliminated in SW620 cells. These findings imply that HKDC1 and the circadian clock interact, as the circadian clock is altered in metastatic cells [156].

Eukaryotic cells adjust to cellular stress by phosphorylating eukaryotic translation initiation factor 2 alpha (eIF2), which results in the translation of specific transcripts that enable the cell to withstand stress [123,124,125,126,157,158]. Activating transcription factor 4 (ATF4) is a leucine zipper transcription factor that modulates the cellular integrated stress response to allow cells to adapt to and endure stressors [133,159,160]. The overexpression of ATF4 causes the HKDC1 gene transcription to increase significantly under cellular stress, changing hepatocyte mitochondrial dynamics [161]. HKDC1 is upregulated in response to the endoplasmic reticulum (ER) stress or mitochondrial respiratory chain inhibition; however, when these stressors are present in combination with RNA interference to decrease ATF4, HKDC1 gene expression is reduced [161].

## 2. Future Directions

Accelerated aerobic glycolysis is a hallmark of cancer cells which provides a rapid source of ATP and good metabolic intermediates for synthesizing nucleic acids, lipids, and proteins in the rapidly dividing cells [162,163]. The increased dependency of cancer cells on glucose metabolism sets them apart from their regular counterparts and could render them more vulnerable to disruption in glucose metabolism. Cancer cells could therefore be selectively targeted through the disruption of glucose metabolism, and the therapeutic targeting of HK enzymes in cancers has seemed to be a plausible strategy. However, considering the overarching redundancy in the catalytic activity of different isozymes, it seems reasonable to argue that one isoform could compensate for another under specified conditions. A lack of literature on the non-redundant functions of each isozyme further complicates this approach. The therapeutic targeting of HKs in cancer per se awaits more targeted approaches for effective outcomes. Identifying isoform-specific roles in cancer could reveal more selective targets that could be utilized for therapeutic purposes without compromising overall homeostasis.

HK1-2 and HKDC1 contain a mitochondrial binding site in the N-terminal domain. This domain mediates HK1 activity in normal cells while it plays a role in tumorigenesis in HK2 and HKDC1 [21,43,164]. HK2 is known to inhibit apoptosis and regulate autophagy [27]. The recent identification of HK2 localization to contact points between mitochondria and endoplasmic reticulum, known as mitochondria-associated membranes (MAMs), has unveiled a novel role of HK2 in regulating Ca^2+^ flux within the cells [165,166]. HKDC1 is also postulated to bind to MAMs similarly and regulate Ca^2+^ flux. In the future, the binding of HK2 and HKDC1 could be specifically targeted as a promising therapeutic strategy for effective outcomes in cancer. Of particular interest, small molecular inhibitors which specifically target the binding of HK2 and HKDC1 to mitochondria and MAMs need further exploration [167]. Such inhibitors have recently been characterized for HK2, which specifically and selectively target HK2 without producing off-target effects. Evaluating similar inhibitors for HKDC1 could prove to be an effective therapeutic avenue for cancer treatment in the future [167]. 

## Figures and Tables

**Figure 1 life-13-00946-f001:**
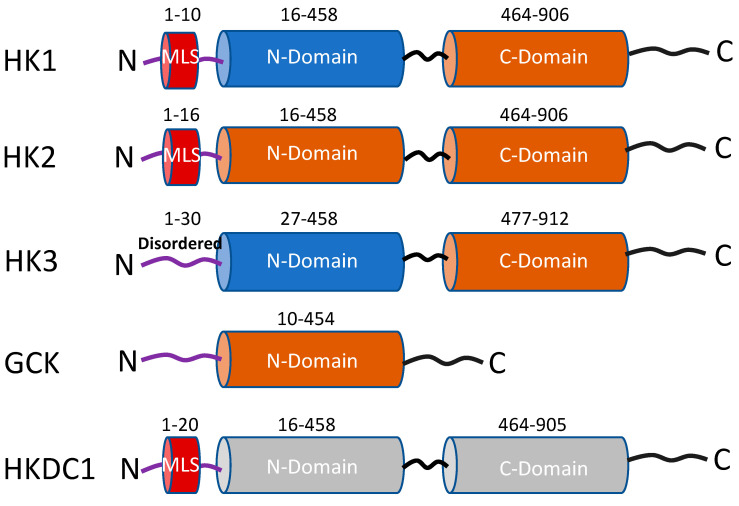
**Schematic representation of the functional domains of the five hexokinase isoforms.** The rust-colored cylinders represent domains with catalytic activity, and the blue cylinders have no catalytic activity. Both cylinders in HKDC1 are gray colored because this isoform has very low kinase activity. MLS = mitochondrial localization sequence (red-colored cylinder). Numbers represent amino acid sequences adapted from Uniprot.org.

**Figure 2 life-13-00946-f002:**
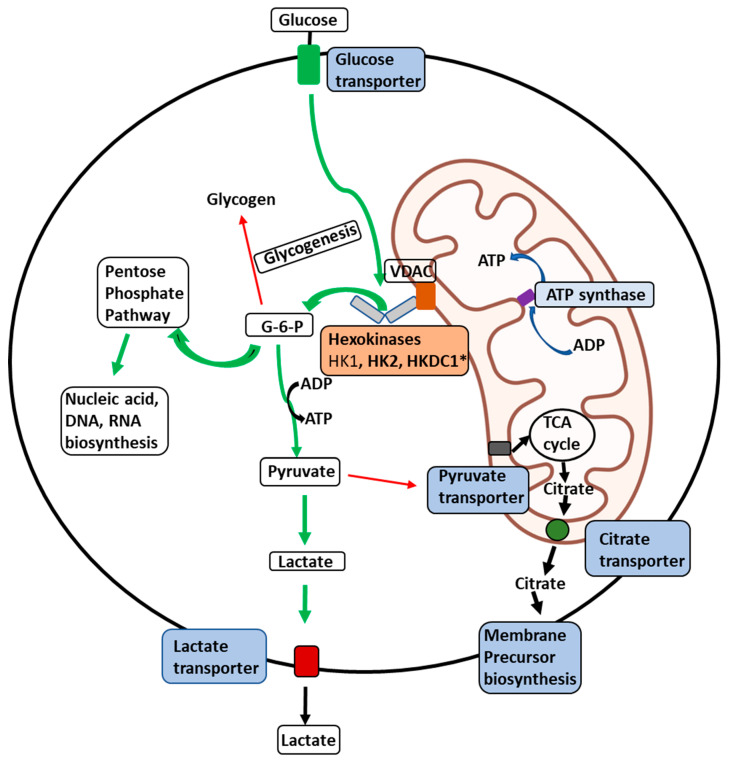
**Illustration of the delivery of glucose to membrane-bound HKs in malignant cells.** Illustration of the glucose delivery to HKs 1, 2, and HKDC1 bound to the outer mitochondrial membrane (OMM) and metabolic fates of the glucose-6-phosphate (G6P) formed thereof within a malignant cell. Glucose transport across the plasma membrane by glucose transporters is phosphorylated by HKs (HK1, HK2, or HKDC1) bound to a voltage-dependent anion channel (VDAC) located on the outer mitochondrial membrane. VDAC allows direct access of ATP generated by the ATP synthase within the mitochondria to the HKs, which can be transported across the inner-mitochondrial membrane by the adenine nucleotide translocator. To maintain malignant cells’ highly glycolytic metabolic flux, the product G6P is rapidly distributed across key metabolic routes (see thick green arrows). The primary metabolic routes for G6P are either entry into the pentose-phosphate pathway for biosynthesis of nucleic acid precursors or conversion to pyruvate and lactate through glycolysis. In cancer cells, most lactate is transported out of the cells with the aid of lactate transporters. In contrast, small amounts of pyruvate are transported to mitochondria through the pyruvate transporters to supply intermediates to the tricarboxylic acid (TCA) cycle (thin red arrows). Citrate transporters transport citrate produced in the TCA cycle to aid in synthesizing membrane components such as phospholipids and cholesterol, essential for tumor cell proliferation. * Novel hexokinase, HKDC1, with roles still unknown.

**Figure 3 life-13-00946-f003:**
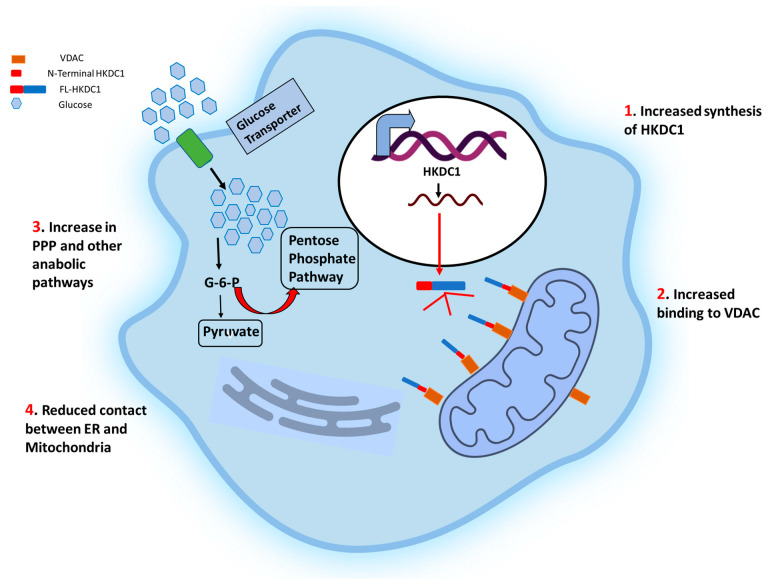
**Schematic representation of the effects of HKDC1 over-expression in cancer cells.** The cell membrane glucose transporters (GLUT 1/3) mediate the glucose uptake, which is degraded to pyruvate by glycolysis. Upregulation of HKDC1 (and other HKs) in many cancer types leads to enhanced generation of glycolytic intermediate, which functions as precursors for numerous metabolic pathways necessary for the biosynthesis of cellular components; pentose phosphate pathway (marked with thick red arrows), cholesterol biosynthesis, and fatty acid biosynthesis. Notably, HKDC1 upregulation leads to an increase in HKDC1-mitochondrial binding, which is responsible for the maintenance of glycolysis and TCA cycle and contributes to unabated cell proliferation through the aversion of apoptosis and endoplasmic reticulum (ER)-mediated stress response mechanisms by reducing the number of physical contact points between ER and mitochondria.

**Table 1 life-13-00946-t001:** Characteristics of HK isoforms in humans.

	HK1	HK2	HK3	GCK	HKDC1
**Gene location (Human)**	10q22	2p13	5q35.2	7p15.1	10q22
**MW (kDa)**	~100	~100	~100	~50	~100
**Number of catalytic domains**	1	2	1	1	1
**Km for glucose** (mmol L^−1^)	0.03	0.3	0.003	6	-
**Km for ATP** (mmol L^−1^)	0.5	0.7	1.0	0.6	-
**G6P inhibition** *K*_i_ (mmol L^−1^)	0.02	0.02	0.10	-	-
**Effect of Pi**	Low conc counteracts G6P inhibition, but high conc is inhibitory	inhibitory	Inhibitory	-	-
**Insulin regulation**	-	+	*	+	*
**Major tissue expression**	Brain, Kidney	Muscle, adipose	Lung, spleen	Liver, pancreas	GI, Kidney, and Brain
**Mitochondrial binding**	✓	✓	✕	✕	✓

+ = effect; - = NO effect; * = Sufficient data not available; Pi = inorganic phosphate; ✓ = binding; ✕ = no binding.

## Data Availability

Data used in this manuscript is publicly available at https://www.cancer.gov/ccg/research/genome-sequencing/tcga accessed on 10 March 2023.

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
