# Peer review of "Aiding Cancer’s “Sweet Tooth”: Role of Hexokinases in Metabolic Reprogramming"

_life, 2023, doi:10.3390/life13040946_

Round 1
Reviewer 1 Report
In this review by Farrooq et al., authors discuss the hexokinases' main features, distribution/expression at the cellular and tissue levels and their roles in cancer-mediated metabolic reprogramming. The manuscript is well written and organized and the topic is of high interest. Although the review provides an updated overview of the role of hexokinases in cancer metabolism and progression, the novelty is partially damaged by more complete reviews on HK2 (PMID: 30543165) but also HKDC1 (e.g., PMID: 34782236). Therefore, the following concerns should be considered to improve the quality and the novelty of this manuscript:
Major comments:
1- The paragraph 1.2 should provide more information about the post-transcriptional regulation of HKs (e.g., miRNA) but also epigenetic regulation (e.g., DNA methylation etc). A table should be added to summarize these regulatory mechanisms.
2- The paragraph 1.2 does not link the alteration of HKs expression in cancers cells with the described regulatory mechanisms. It is important to understand for each cancer, why HKs are deregulated and thus to propose potential therapeutic strategies (in another paragraph). This can be also helpful to identify common regulatory mechanisms between cancer types. A table and/or a figure documenting these alterations should be added.
3- The review should provide a comparison of HKs expression between cancer types. Underlying the most relevant HKs/cancer type will provide important insights to discuss potential therapeutic approaches.
4- A section on the regulation/trafficking of hexokinases should be inserted.Recent data point out that HK can be post-transcriptionally modified and shuttle between cells by extracellular vesicles.
5- The paragraph 1.4 is not clear. More precisions should be added to understand if the described mechanisms are for healthy or pathological conditions. Is there any difference of HKs localization in cancer cells? Any difference between cancer types?
6- The review should include a section dedicated to the therapeutic strategies aiming at targeting HKs in cancers. Such paragraph should discuss the potential inhibitors (e.g., HK2 inhibitors: PMID: 36278443) but also the potential strategies targeting the regulatory mechanisms of HKs expression. A table should summarize the different possibilities with examples.
6- Some references should be added and discussed:
- PMID 36736415 about the uncoupling of HK2 from the mitochondria
- PMID 36804531 about the regulation of HK by lncRNAs
- PMID 35305311 about the HK1 localization
Minor comments:
1- Line 374: please remove one dot before “A lack of literature…”
2- Table 1 should also indicate the Km for fructose. Any inhibition by fructose-6-P?
3- The MS should be carefully verified to correct minor typos
Author Response
Major comments:
- The paragraph 1.2 should provide more information about the post-transcriptional regulation of HKs (e.g., miRNA) but also epigenetic regulation (e.g., DNA methylation etc). A table should be added to summarize these regulatory mechanisms.
Response: Thank you for pointing this out to allow us to improve this manuscript. As per the suggestion, more information has been included in this section regarding the role of microRNAs and DNA methylation in regulating hexokinase expression. However, it should be noted that most of these non-canonical transcriptional regulatory mechanisms, like microRNAs and DNA methylation, have been described in various cancers concerning HK2, which is the most highly expressed HK in cancers. Therefore, an apparent lack of similar data on other HKs in this manuscript is due to a lack of such data in the existing literature.
- The paragraph 1.2 does not link the alteration of HKs expression in cancers cells with the described regulatory mechanisms. It is important to understand for each cancer, why HKs are deregulated and thus to propose potential therapeutic strategies (in another paragraph). This can be also helpful to identify common regulatory mechanisms between cancer types. A table and/or a figure documenting these alterations should be added.
Response: Thank you for bringing this to our notice. More information has been included to elaborate on the regulation of hexokinases under normal and tumorigenic conditions. Specifically, a detailed section has been included to discuss the regulation of HK2 in various forms of cancer and its role and the possible role of its upstream effector Akt in the regulation of HKDC1. This new information provides valuable inputs into non-overlapping functions of HKs in cancer, providing opportunities for therapeutic interventions to target specific isoforms.
- The review should provide a comparison of HKs expression between cancer types. Underlying the most relevant HKs/cancer type will provide important insights to discuss potential therapeutic approaches.
Response: Thank you for this valuable comment. We have done additional data mining from the TCGA, and a figure (Supplementary Fig 2) has been added to the revised manuscript as advised, that shows the levels of HK isoforms in different cancer types.
- A section on the regulation/trafficking of hexokinases should be inserted. Recent data point out that HK can be post-transcriptionally modified and shuttle between cells by extracellular vesicles.
Response: We are grateful to the reviewer for this suggestion and more information in the section on the subcellular localization of HKs has been included in the revised manuscript.
- The paragraph 1.4 is not clear. More precisions should be added to understand if the described mechanisms are for healthy or pathological conditions. Is there any difference of HKs localization in cancer cells? Any difference between cancer types?
Response: We apologize for this confusion. We have clarified the statements on HK localization depending on whether the statement refers to normal or cancer cells. The literature has not explored differences in subcellular localization for each cancer type. Therefore, we have discussed differences in subcellular localization in normal versus cancer cells, especially concerning HK1 and HK2, by providing a broad overview of the typical subcellular localization of these HKs in normal versus cancer cells, along with the possible role of altered localization under tumorigenic conditions.
- The review should include a section dedicated to the therapeutic strategies aiming at targeting HKs in cancers. Such paragraph should discuss the potential inhibitors (e.g., HK2 inhibitors: PMID: 36278443) but also the potential strategies targeting the regulatory mechanisms of HKs expression. A table should summarize the different possibilities with examples.
Response: The role of small molecular inhibitors, such as HK2 inhibitors described in the previous paper (PMID: 36278443) on HK2 regulation and therapeutic targeting has been discussed in the “discussion” section. A small section on the role of potential strategies targeting the regulatory mechanisms of HKs expression, for example, the regulation of HKDC1 by HK2, has also been briefly discussed.
6- Some references should be added and discussed:
- PMID 36736415 about the uncoupling of HK2 from the mitochondria
Response: This has been added to the revised manuscript.
- PMID 36804531 about the regulation of HK by lncRNAs,
Response: This has been added to the revised manuscript.
- PMID 35305311 about the HK1 localization
Response: This has been added to the revised manuscript.
Minor comments:
- Line 374: please remove one dot before “A lack of literature…”
Response: This has been edited in the revised manuscript.
- Table 1 should also indicate the Km for fructose. Any inhibition by fructose-6-P?
Response: The Km for fructose has not been reported in the papers evaluating the kinetics of HKs.
- The MS should be carefully verified to correct minor typos
Response: The entire has been thoroughly proofread as advised.
Reviewer 2 Report
Farooq Z. and colleagues present a concise review on hexokinases and their crucial role in cancer metabolic reprogramming. The review provides a helpful summary and overview for readers in the related field. However, the following critiques and comments need to be addressed:
1, Table 1 needs clearer annotation. For instance, what do the numbers (0.02, 0.002, 0.10) mean in G6P inhibition? What does "Effect of pi" mean? Does it refer to the effect of phosphate on HK activity? If so, please specify. Also, indicate that "pi" means "phosphate" in the table. The table contains various symbols (+, -, *, and v), but only - and * are annotated. Please annotate + and v as well. What does + indicate for G6P inhibition and insulin regulation, respectively? Finally, regarding mitochondrial binding, some studies suggest that GCK also binds to mitochondria. The author should clarify this point and include the discussion here, citing references such as Sternisha SM, Miller BG. Molecular and cellular regulation of human glucokinase. Arch Biochem Biophys. 2019 Mar 15;663:199-213. doi: 10.1016/j.abb.2019.01.011. Epub 2019 Jan 11. PMID: 30641049; PMCID: PMC6377845. and Lee JW, Kim WH, Lim JH, Song EH, Song J, Choi KY, Jung MH. Mitochondrial dysfunction: glucokinase downregulation lowers interaction of glucokinase with mitochondria, resulting in apoptosis of pancreatic beta-cells. Cell Signal. 2009 Jan;21(1):69-78. doi: 10.1016/j.cellsig.2008.09.015. Epub 2008 Oct 7. PMID: 18940247.
2, In Figure 1, in addition to the schematic graph, it would be helpful to include a protein alignment analysis for the five different isoforms.
3, In lines 386-387, to support the claim that blocking the binding of HK2 and KHDC1 could be a promising therapeutic target, more references and discussion should be provided. What studies have demonstrated that blocking such binding could lead to therapeutic benefits?
4, The manuscript contains many grammatical errors that need to be corrected. For example, in line 82, GK instead of GCK is used for glucokinase. Always introduce abbreviations when the term is first mentioned. In line 102, VDAC should be given the full name. There is no need to repeatedly use both the full and abbreviated terms afterwards. In lines 99-100, "eyes'" and "retina" are duplicated. Please carefully check for other errors and correct them accordingly.
Author Response
1, Table 1 needs clearer annotation. For instance, what do the numbers (0.02, 0.002, 0.10) mean in G6P inhibition? What does "Effect of pi" mean? Does it refer to the effect of phosphate on HK activity? If so, please specify. Also, indicate that "pi" means "phosphate" in the table. The table contains various symbols (+, -, *, and v), but only - and * are annotated. Please annotate + and v as well. What does + indicate for G6P inhibition and insulin regulation, respectively? Finally, regarding mitochondrial binding, some studies suggest that GCK also binds to mitochondria. The author should clarify this point and include the discussion here, citing references such as Sternisha SM, Miller BG. Molecular and cellular regulation of human glucokinase. Arch Biochem Biophys. 2019 Mar 15;663:199-213. doi: 10.1016/j.abb.2019.01.011. Epub 2019 Jan 11. PMID: 30641049; PMCID: PMC6377845. and Lee JW, Kim WH, Lim JH, Song EH, Song J, Choi KY, Jung MH. Mitochondrial dysfunction: glucokinase downregulation lowers interaction of glucokinase with mitochondria, resulting in apoptosis of pancreatic beta-cells. Cell Signal. 2009 Jan;21(1):69-78. doi: 10.1016/j.cellsig.2008.09.015. Epub 2008 Oct 7. PMID: 18940247.
Response: Thank you very much for bringing this to our attention. As advised, Table 1 has been properly annotated in the revised manuscript.
Reports in the literature suggest that GCK also localizes to mitochondria. However, GCK lacks the characteristic N-Terminal mitochondrial binding domain (MBD), which remains the central discussion throughout this manuscript. Also, GCK expression is limited to the pancreas and liver. For these reasons and also due to a limited number of papers on the role of GCK in cancer, this aspect of GCK has not been discussed in this review. However, in lines 85-90, we have touched base on the role of mitochondrial localization of GCK.
2, In Figure 1, in addition to the schematic graph, it would be helpful to include a protein alignment analysis for the five different isoforms.
Response: As advised, we have performed a protein alignment analysis for the five different isoforms, and we are submitting a percent identity matrix (Supplementary Fig 1) to the revised manuscript.
3, In lines 386-387, to support the claim that blocking the binding of HK2 and KHDC1 could be a promising therapeutic target, more references and discussion should be provided. What studies have demonstrated that blocking such binding could lead to therapeutic benefits?
Response: This has been included in the discussion in the revised manuscript (lines 430-438)
4, The manuscript contains many grammatical errors that need to be corrected. For example, in line 82, GK instead of GCK is used for glucokinase. Always introduce abbreviations when the term is first mentioned. In line 102, VDAC should be given the full name. There is no need to repeatedly use both the full and abbreviated terms afterwards. In lines 99-100, "eyes'" and "retina" are duplicated. Please carefully check for other errors and correct them accordingly.
Response: Thank you for this careful observation. Typographical and grammatical errors have been corrected in the revised manuscript.
Reviewer 3 Report
Authors of this manuscript provided a concise review of hexokinases enzymes regarding their functions, regulations, and relevance to diseases such as cancer. The manuscript is well-written and easy to follow. I have a few suggestions for the authors to improve the manuscript. They are listed in the following.
11. Line 56, should be “Amino acids sequence”.
2. Page 2, line 60, what is the rate of amino acid substitution? Should be it rate of mutation through evolution?
3. Page 2, line 61, which species of HK1 was referred here?
4. Page 2, line 85, which enzyme is localized in the cytoplasm?
5. Line 99, the comma after “eyes” needs to be corrected.
6. Table 1, references to the Km data, tissue expression of HKs should be provided. Also, what is “Effect of pi”?
7. Line 137, transcriptional upregulation of HK2 by what?
8. Line 138, should be “high levels of epigenetic marks like”.
9. Line 153, the period after HK2 should be removed.
10. Line 156, the “l” should be capitalized.
11. Line 162, should be “Lowest affinity”.
12. Line 164, mixed use of G6P and G-6-P were found. The notation of “G6P” should be consistent.
13. Line 185, it is unclear what are the kinetic properties of HKDC1? Kinetic of glucose metabolism by HKDC1 or something else?
14. Line 191, remove “some”.
15. Line 238, which tumor models?
16. Line 288, what is activating variations? Was it “17 activating mutations”?
17. Lines 293-294, should it be somatic mutations?
Author Response
Authors of this manuscript provided a concise review of hexokinases enzymes regarding their functions, regulations, and relevance to diseases such as cancer. The manuscript is well-written and easy to follow. I have a few suggestions for the authors to improve the manuscript. They are listed in the following.
- Line 56, should be “Amino acids sequence”.
Response: The error has been corrected in the revised manuscript.
- Page 2, line 60, what is the rate of amino acid substitution? Should be it rate of mutation through evolution?
Response: That is correct. The statement” rate of mutation through evolution” has been included in the revised manuscript for clarity.
- Page 2, line 61, which species of HK1 was referred here?
Response: It is humans. The term has been included in Table 1 for clarity in the revised manuscript.
- Page 2, line 85, which enzyme is localized in the cytoplasm?
Response: It refers to glucokinase (GCK).We have replaced “the enzyme” with “GCK” for clarity in the revised manuscript.
- Line 99, the comma after “eyes” needs to be corrected.
Response: Done as advised.
- Table 1, references to the Km data, tissue expression of HKs should be provided. Also, what is “Effect of pi”?
Response: This information has been included in Table 1 in the revised manuscript.
- Line 137, transcriptional upregulation of HK2 by what?
Response: It indicates transcriptional upregulation by insulin since the statement refers to presence of insulin response element in the promoter of HK2. However, the term “insulin” has been included in the in the revised manuscript for clarification.
- Line 138, should be “high levels of epigenetic marks like”.
Response: Corrected as advised.
- Line 153, the period after HK2 should be removed.
Response: Corrected as advised.
- Line 156, the “l” should be capitalized.
Response: Corrected as advised.
- Line 162, should be “Lowest affinity”.
Response: Corrected as advised.
- Line 164, mixed use of G6P and G-6-P were found. The notation of “G6P” should be consistent.
Response: Corrected as advised.
- Line 185, it is unclear what are the kinetic properties of HKDC1? Kinetic of glucose metabolism by HKDC1 or something else?
Response: Indeed, it refers to the kinetics of glucose phosphorylation by HKDC1.
- Line 191, remove “some”.
Response: Corrected as advised.
- Line 238, which tumor models?
Response: These models are described in the references mentioned at the end of the statement.
- Line 288, what is activating variations? Was it “17 activating mutations”?
Response: That is corrected, The word “activating variations” has been replaced with “activating mutations” for clarity in the revised manuscript.
- Lines 293-294, should it be somatic mutations?
Response: Corrected as advised. The word “somatic variations” has been replaced with “somatic mutations” for clarity in the revised manuscript.
Round 2
Reviewer 2 Report
The revision looks fine to me
Reviewer 3 Report
Authors have addressed all issues by this reviewer.